# Spatial and Temporal Interaction Coupling of Digital Economy, New-Type Urbanization and Land Ecology and Spatial Effects Identification: A Study of the Yangtze River Delta

**Yuqi Zhu** [1,2,3,4], **Siwei Shen** [1,4], **Linyu Du** [1,4], **Jun Fu** [1,3], **Jian Zou** [1,3], **Lina Peng** [1,3] and **Rui Ding** [1,2,3,4,*]

1    College of Big Data Application and Economics (Guiyang College of Big Data Finance), Guizhou University of Finance and Economics, Guiyang 550025, China
2    Guizhou Key Laboratory of Big Data Statistical Analysis, Guizhou University of Finance and Economics, Guiyang 550025, China
3    Guizhou Collaborative Innovation Center of Green Finance and Ecological Environment Protection, Guiyang 550025, China
4    Artificial Intelligence and Digital Finance Lab, Guizhou University of Finance and Economics, Guiyang 550025, China
*    Correspondence: 201801162@mail.gufe.edu.cn

**Abstract:** In the digital era, the contradiction between regional urban development and land ecological protection is still prominent. Clarifying the relationship and internal interaction logic among digital economy (DE), new-type urbanization (NU), and land ecology (LE) is of great significance to the region's sustainable development. Based on theoretical analysis, this study examines the relationship among DE, NU, and LE in the Yangtze River Delta through spatial analysis and empirical test with the city data from 2011 to 2020. The study found that: (1) The overall development level of DE–NU–LE in the Yangtze River Delta shows a steady upward trend, the development level of DE and NU lags behind LE, and the convergence trend among them gradually strengthened. (2) The DE-NE-LE and the coupling coordination have different and complex spatial and temporal dynamic evolution characteristics. The ability for coordinated development is enhanced continuously, which presents a typical pattern of "high in the east and low in the west". (3) The DE has a lasting role in promoting the development of the NU and LE, while the support and stimulation of NU and LE for DE needs to be strengthened. The relationship between the NU and LE shows a mutually restricted trend. (4) The DE has a significant "siphon effect". While NU and LE both have significant positive spatial spillover effects, which can promote the coordinated development of surrounding cities. This study deepens the understanding of DE–NU–LE coordinated development, and provides a new perspective for sustainable urban development and alleviating land conflicts.

**Keywords:** land ecology; digital economy; urbanization; land conflict; coupling coordination

## 1. Introduction

Land is one of the most important foundations for human survival and development and is an irreplaceable material wealth of human society. The healthy and safe development of LE is not only the environmental foundation for regional development but also an important guarantee for sustainable development. However, since the industrial revolution, with accelerated urbanization and industrialization, the social and economic development has been rapid. Land resources development and protection have been greatly impacted and affected, and the competition for ecological space has been frequent. In this context, the contradiction between humans and land caused by rapid economic development and urbanization has become increasingly prominent, and regional sustainable development is facing severe challenges [1]. As China's development enters a new era, the construction of ecological civilization is elevated to a national strategy. How to coordinate the development

of LE and urbanization is a popular proposition in current urban development. To this end, *the 14th Five-Year Plan for the Implementation of New-Type Urbanization* emphasizes the comprehensive planning of national land space. The ecological protection red lines and urban development boundaries have been defined and implemented to improve the quality and stability of the ecosystem. At the same time, digital technology application scenarios are enriched to improve urban governance and land management. Similarly, *the National Adaptation Strategy for Climate Change 2035* mentions promoting the construction of digital twins for the river basins. Additionally, the digital platform will be established for ecological governance technologies to strengthen early warning monitoring and emergency response capabilities for water and land resources. It can be seen that the DE and its related technologies have become an important driving force for the high-quality development of Chinese cities. Therefore, coordinating the relationship among the development of DE, NU, and the protection of LE has great practical significance for sustainable regional development and the construction of ecological civilization.

The related studies around DE, NU, and LE have received extensive attention from scholars. Most current literature only discusses the development and influence of a single system, and the research on the relative relationship between multiple systems is weak. Especially, less attention is paid to the coupling mechanism and coordinated development among the DE, NU, and LE. First of all, from the related research on DE, the DE concept has not yet been unified [2]. The term "digital economy" can be traced back to 1996, when it was first proposed by Tapscott Don [3], an American businessman. For a long time, scholars have further discussed and enriched the concept and connotation of the DE. As Lane [4] pointed out, the DE refers to the convergence of computing and information technology on the Internet and the resulting flow of information technology has triggered the electronic transformation of business. Later, Moulton [5] believed that the information technology industry and its related technologies, digital transactions of goods and services, and tangible commerce, retail, and related investments supported by digital technologies are all part of the DE. Then, Dahlman [6] mentioned that the DE is the amalgamation of economic and social activities with digital information as the main production factor and supported by information technologies such as big data and cloud computing.

In terms of social development, the DE plays a multi-tiered driving role. On the one hand, the DE can promote benign adjustment at the micro level of cities by improving resource misallocation [7], enhancing urban total factor productivity, and promoting industrial innovation [8,9]. On the other hand, the DE also plays a positive role in macro-environmental development such as green and low-carbon development [10], urban and rural coordination [11], inclusive growth [12,13], and economic and financial inclusion [14,15].

Second, from the research related to NU, the term "new-type urbanization" originates from China and is an urbanization path with distinctive Chinese characteristics. Hence, the current research on NU is also basically focused on China. Compared with traditional urbanization, which is mainly based on land development, the NU, which upholds the core connotation of being people-oriented is more intensive, efficient, harmonious, and sustainable [16]. Similarly, the influence of NU is also multi-faceted for economic and social development. The development concepts of green, low-carbon, and recycling are conducive to reducing environmental pollution [17], and improving energy utilization efficiency [18,19]. The NU shows significant ecological benefits [20], which accelerates urban green transformation. The core connotation of being people-oriented, puts more emphasis on human development and has greater inclusiveness and universality, which facilitates the stable transfer of rural labor and improves the income level of rural residents [21]. Under the overall framework of urban-rural integration development, bridging the income gap between urban and rural is conducive to accelerating rural revitalization and enhancing the happiness of urban and rural residents [22,23]. However, it is worth noting that although NU has the ecological effect of "reducing pollution and increasing efficiency", it also poses a great challenge to land resources. Against the background of decreasing cultivated land, inputs of high-carbon production materials such as chemical

fertilizers and pesticides are on the rise to ensure food production. The overuse of chemical fertilizers can cause soil acidification [24], nutrient imbalance in soils, and destroy soil biodiversity [25], which puts pressure on the sustainability of the land. Although pesticides can prevent and mitigate the effects of pests, they also further amplify the risk of soil and water pollution [26]. In addition, large-scale infrastructure construction tends to cause some land ecological problems such as soil degradation and erosion [27], etc. Therefore, how to find a balance between the construction of NU and the protection of LE is a real problem for the current sustainable development of the region.

Finally, from the relevant studies on LE, the land ecosystem is a complex dynamic system formed by the interaction of interdependent natural elements and human activities on the Earth's surface [28]. From the above definition, first of all, land ecosystem includes the continuous distribution of elements on the Earth's surface, such as agricultural land, grassland, forest, etc., which are the main bearers of resources [29]. As part of the ecosystem, land ecology is more focused on the condition of the land. It is the basis for land use planning and structural layout. Secondly, the LE considers the human–land relationship and the derived ecological values [30]. Through reasonable input and transformation of the land, it can cause LE to develop in a more efficient direction and to obtain the corresponding economic and living benefits [31,32]. Conversely, predatory human activities on the land can lead to ecological degradation of the land [33]. Finally, the LE is an open and dynamic system, and its condition is not static. On the one hand, the LE can resist the adverse changes of the environment with the self-regulating and compensatory effects [34]. However, its effectiveness is limited. On the other hand, external shocks such as biological activities, landscape transformation, climate change, and natural disasters can have certain impacts on LE. The sustainable development of land resource use and social economy are further affected. From the relevant studies, compared with studies on the ecological environment, existing research on LE is limited. Researches on LE mainly focus on land ecological security assessment [35,36], land ecological efficiency [37], land resources [33,38], land use [39], and land sustainable development [40]. For instance, Feng [41] systematically assessed the land ecological safety in Ningbo, a coastal city in eastern China, and summarized its temporal characteristics. Then Yu [42] used the slacks-based measure-undesirable (SBM-UN) model to study the land use efficiency of urban agglomerations in China and further analyzed its driving factors. In addition, some scholars have also studied the relationship between urbanization and land. Although the conclusions are different, the contradiction between urbanization and land is particularly common [40,43,44].

To sum up, the synergistic development of DE, NU, and LE is a necessary path for Chinese cities to move toward high-quality development. Nevertheless, the development foundations of DE, NU, and LE in Chinese cities are quite different at present, and the development levels are uneven. Furthermore, the multi-system coordination mechanism is not yet clear. Given this, this paper takes the Yangtze River Delta as a study area to explore the coordinated development mechanism of subsystems. The spatial and temporal evolution characteristics of subsystems coupling coordination development are revealed by the coupling coordination model, which is modified. Finally, the interaction mechanism and spatial effects of subsystems are identified with the panel vector autoregression (PVAR) model and spatial econometric model. The contributions and innovations of this paper mainly lie in the following three aspects: (I) In terms of theoretical innovation, this paper incorporates DE, NU, and LE into a unified analytical framework and systematically discusses the internal logic and coupling mechanism of subsystems. The relevant achievements can be enriched, which fills a partial vacancy in the research on the linkage development mechanism of DE, NU, and LE. (II) In terms of empirical evidence, this paper reveals the different evolutionary characteristics of DE, NU, and LE in the Yangtze River Delta and further explains the reasons behind them. It clarifies the spatial and temporal evolution characteristics of the subsystems and the coupling coordination development. (III) In terms of path innovation, this paper explores the spatial effect of DE, NU, and LE,

and provides a spatial perspective for promoting the coordinated development of DE, NU, and LE.

The rest of the sections in this paper are arranged as follows: Section 2 is the portrayal of the coupling mechanism. Section 3 is the introduction of the study area, methods, and data resources. Section 4 is the analysis of the empirical results. Section 5 is the identification of the interaction mechanisms and spatial effects. Section 6 is the conclusions and suggestions.

## 2. The Coupling Mechanism

Based on the existing studies, this paper constructs a mechanism analysis framework for the coupling of DE, NU, and LE according to the interaction relationship among the three (Figure 1).

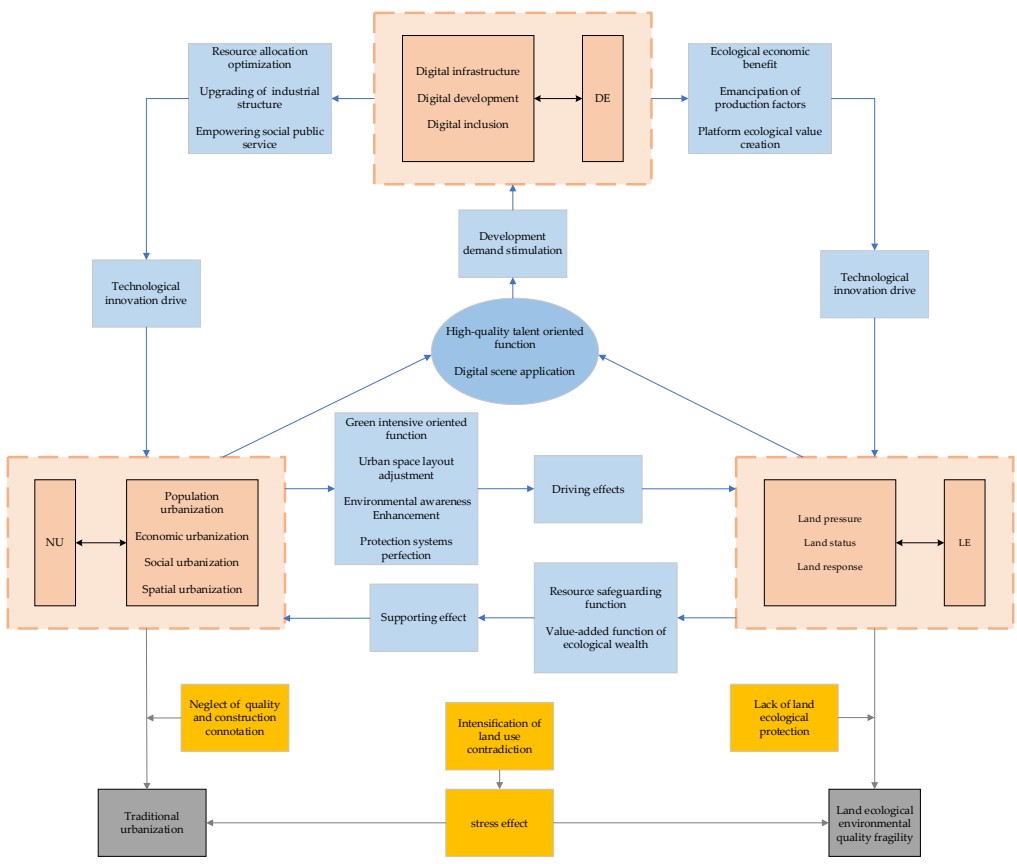

**Figure 1.** The coupling mechanism of DE–NU–LE.

Science and technology are the first productive forces of socio-economic development. The DE, mainly supported by information technologies such as artificial intelligence, big data, and the Internet [45], presents obvious new technological characteristics. The resulting series of technological and industrial innovations has become a powerful driving force for the development of NU and LE. Specifically, the driving effect of DE on NU is mainly manifested in three aspects: (I) The DE has an optimization effect on resource allocation [7]. Digital technology has changed the traditional method of resource allocation. The industrial changes and technological innovation in cities are effectively promoted by rapid flow and utilization efficiency improvement of production resources. It accelerates local urbanization [46] and enhance urbanization efficiency [47]. (II) The DE facilitates industrial upgrading and provides new options for bridging the income gap. The technological advantage of DE further generates new industries and transforms traditional industries to provide employment and training opportunities for more people [10,48]. The residents' income will rise because of working, which could reduce the urban-rural income gap [11]

and enhance residents' well-being [49]. The experience of developed regions shows that the rational flow of resource factors and increase in employment will be more conducive to improving the quality of urbanization [50]. (III) Digital technology and its applications can empower the construction of social public services and provide a new path to achieve equalization of public services [51]. The improvement of public service quality is the core content of NU, its improvements such as medical care, transportation, and education plays an important role in promoting the happiness of residents' lives. The development of DE is conducive to accelerating the digital transformation of urban public services, improving the efficiency of public affairs, and enhancing the credibility of government data. Meanwhile, high-quality resources in healthcare and education can be digitally disseminated. The process of equalization of public services is accelerated through digital dissemination to cover and influence more places and people.

In terms of LE, there are good performances of DE in enhancing the ecological quality of land: (I) the ecological and economic benefits that the DE itself possesses. On the one hand, the DE has good ecological benefits in the process of promoting the transformation of traditional industry. It can effectively reduce industrial pollution emissions [52], improve energy efficiency [53], and achieve low-carbon transformation [54], which plays an important role in the overall protection of the ecological environment. On the other hand, the DE can effectively promote economic growth [55], thus fully guaranteeing the investment of funds for ecological protection and restoration. (II) The emancipation effect of DE on production factors. Firstly, the DE uses digital information as the main production factor, which has the characteristics of being clean, intensive, and efficient [10]. Secondly, the DE can reduce the cost of land production factors and improve agricultural efficiency [56]. The application of digital technology in the field of agricultural production can achieve savings in labor, water, land, and fertilizers [57,58]. The digital liberation of agricultural productivity provides greater scope for replenishing degraded land, enhancing land resilience, and achieving rational landscape planning and management [59]. (III) The DE has the role of platform ecological value creation [60]. On the one hand, the government-led digital platform can form a closely linked ecological collaborative governance model; and achieving overall regional ecological protection and rational planning through digital monitoring, management, and precise governance. On the other hand, an enterprise and market-led platform can effectively utilize the timeliness and convenience of the information network. The land ecological value can be increased by transformation of land productivity to economic benefits on platforms such as e-commerce. Accordingly, NU and LEs will continue to attract high quality talent for their own development needs, which accumulates human capital for the development of DE. With the in-depth development of NU and LE, the demand for quality improvement will reverse stimulate the development of related technologies [61]; also creating new development opportunities and application scenarios for digital development. The application of digital ecological governance, digital government, digital twin cities, and other related technologies is not only a result of the development of the DE itself, but also an inherent requirement of NU and LE.

In terms of the interaction between NU and LE, the driving effect of NU on LE is mainly reflected in three aspects: (I) The NU has intensive green oriented function [16]. The green, intensive, intelligent, and sustainable development concept of NU places more emphasis on the coordinated development between cities and the ecological environment rather than mutual trade-offs [37]. Under the guidance of relevant concepts, the predatory land management mode is gradually abandoned. Through the active guidance of the government, the industrial development model with high pollution, high emissions, and high consumption is in a transform position. Eventually, the intensive use of land is realized through the transformation of urban industry and operation mode. (II) The NU puts forward more strict and precise requirements on urban spatial layout. It clearly limits the urban development boundary, and the restoration space can be retained for land. (III) The continuous progress of NU is accompanied by the improvement of living standards. The individual and collective awareness of environment protection is strengthening with

rapid growth of public demand for high-quality urban space and land security. The government also strengthens the protection of LE and accelerates the perfection of the relevant protection systems in order to meet the growing ecological needs of the public.

A good land ecological environment is the foundation for the smooth progress of NU. It is mainly reflected in two aspects: (I) The LE has the function of resource guarantee [29]. Land is scarce, and land that can be used for production and housing is even scarcer. In order to realize the sustainable development of land resources, the protection of LE is needed. The safe development of LE, especially food security, is the material guarantee and spatial basis for human survival and urban development. (II) The LE has a wealth-adding function [62]. It provides new possibilities for dealing with the problem between urban development and land conservation. Firstly, through the marketization and valorization of land productivity, especially agricultural products, the advantages of land resources are converted into ecological product advantages. Secondly, the accumulation of land ecological assets is realized through ecological compensation and green financial products such as carbon finance. Ultimately, the mutual integration of ecological capital and urban development can be realized. Therefore, NU and LE complement each other and their coordinated development is an effective way to achieve sustainable development. However, due to the lesser attention paid to the quality of development and the core meaning of the NU, urban construction is prone to take the old path of being extensive. The over-exploitation of land resources and the growing demand for land will further squeeze the ecological space of land, intensify land conflicts, and even form a mutual stress effect [43,44].

## 3. Study Area, Materials and Methods

### 3.1. Study Area and Data Sources

According to *the outline of the integrated regional development of the Yangtze River Delta* issued in 2019, the scope of the Yangtze River Delta includes Jiangsu Province, Zhejiang Province, Anhui Province, and Shanghai City, with a total of 41 cities in the whole area. As of the end of 2020, the Yangtze River Delta accounts for 16.79% of the country's population and produces 24.09% of the country's GDP on 3.7% of the country's land area. At present, the Yangtze River Delta has become one of the regions with the most developed economic development in China. It is an important leading area and demonstration area for the development of DE, the construction of NU, and ecological civilization in China. The Yangtze River Delta region has modern port groups and airport groups, high density traffic arteries, basic connectivity of major infrastructure, and relatively balanced development of public services. The interactive mechanism of ecological protection and the pattern of coordinated urban and rural development are gradually taking shape. In terms of DE development, the Yangtze River Delta region has obvious advantages in technological innovation. In 2020, the total digital economy will reach CNY 10.83 trillion, about USD 1.57 trillion, accounting for 44.26% of the Yangtze River Delta's GDP, and its development level ranks on the top-tier for China.

The data used in this paper are mainly from *the China City Statistical Yearbook*, *China City Construction Yearbook*, provincial and municipal statistical yearbooks, local statistical bulletins, and the EPS database. The PM2.5 data are obtained from the Surface PM2.5 Dataset of the Atmospheric Composition Analysis Group at Washington University in St. Louis. The data on digital financials are obtained from the China Digital Finance Inclusion Index measured by the Digital Finance Research Center of Peking University in cooperation with Ant Financial Services Group (https://www.dfor.org.cn/, accessed on 19 January 2023). The missing values of the necessary data are supplemented by linear average and interpolation.

*3.2. Study Methods*

3.2.1. Indicator System Construction

Based on the discussion of the coupling mechanism above, this paper makes a comprehensive evaluation of DE, NU, and LE from multiple dimensions. The specific construction methods and indicators are as follows.

The DE, as a new economic form with data resources as production factors and digital technology as the main mode of production, has obvious technical characteristics. Additionally, it is based on information technology, with Internet technology as the carrier. In terms of measurement, most scholars currently select representative indicators from different dimensions to reflect its information construction and the foundation of the Internet, and the development of related digital industries [55,63]. Therefore, following the ideas mentioned above, this paper makes a comprehensive assessment of the DE development from three dimensions: digital infrastructure, digital development, and digital inclusion, where digital infrastructure can reflect the material conditions and market potential of digital development, and usually be measured by internet penetration rate and mobile phone usage. Digital development can represent the development vitality of the DE, which is mainly measured by digital practitioners and telecommunications business volume. Additionally, digital inclusion reflects the degree of integration between digital technology and social economy, which is measured by digital finance inclusion index.

The NU, with distinctive Chinese characteristics, is an inheritance and abandonment of traditional urbanization. It aims to achieve comprehensive human development and attaches importance to the coordinated development of multi-dimensions such as economy, society, space, and population. The construction of an urbanization evaluation system in existing studies has also changed from a single evaluation index measured by urbanization rate to a multi-dimensional comprehensive evaluation system. Additionally, the main features and basic connotations of the NU have been highlighted. Therefore, drawing on the existing research [27,64], and considering the availability and repetitiveness of data, the urbanization of population-economy-society-space is used as the evaluation basis of NU development, where population urbanization emphasizes changes in resident attributes, which mainly include urbanization rate and urban employed population. Economic urbanization is intended to represent the potential of urban economic development, including economic growth rate, residents' income, and the proportion of modern industrial output. Social urbanization emphasizes the improvement of the city's social environment and the popularization of public services, such as public transport, expenditure of science, education, air quality, and social consumption level. Spatial urbanization represents the spatial basis of urban development, which is usually measured by the expansion of built-up areas and space per capita.

As an important part of the ecological environment, The LE puts more emphasis on land carrying capacity, land structure, and land function. At present, there are two main evaluation methods for the LE. One is to start from economic and social benefits, consider creating more economic value with less environmental sacrifice, and use land ecological efficiency to measure the development of LE. The other is to use the index system method to evaluate LE from different dimensions. Xu (2014) evaluated Guangzhou's land ecological security from demand and supply aspects. Chen [31] made a fuzzy assessment of China's land resources based on the pressure-state-response (PSR) framework. Considering the availability and accuracy of prefecture-level city data, this paper conducted a comprehensive assessment of LE in the Yangtze River Delta using the PSR framework. Due to the scarcity of land, the pressure on land mainly comes from population growth, industry development, and land pollution. Therefore, urban population density, agricultural economic efficiency, and land pollutants are used to represent the degree of land pressure. The land status is a comprehensive consideration of land structure, land function, and overall development, which is usually measured by industrial land, green area, food security, soil and water coordination degree, land economic density, and the green development index. The response indicators are positive feedbacks and concerns to changes of LE [35].

Generally, cities with better economic development pay more attention to the governance and protection of the ecological environment. Thus, GDP per capita, pollutant treatment rate, and green coverage area that are a general representation of land restoration are used to measure the land response.

The specific index system is shown in Table 1.

**Table 1.** Evaluation indicators and weights of DE–NU–LE development in the Yangtze River Delta.

| Subsystem Layer | Dimensional Layer | Indicator Layer | CRITI Weights | Entropy Weights | Type |
|---|---|---|---|---|---|
| DE | Digital infrastructure | Number of Internet users per 10,000 people | 0.0916 | 0.1776 | + |
| | | Number of cell phone subscribers per 10,000 people | 0.0808 | 0.1743 | + |
| | Digital development | Computer services and software practitioners | 0.5050 | 0.2711 | + |
| | | Total telecommunications business | 0.2452 | 0.2047 | + |
| | Digital inclusion | Digital finance inclusion index | 0.0775 | 0.1721 | + |
| NU | Population urbanization | Urbanization rate | 0.0464 | 0.0870 | + |
| | | The proportion of employees in the secondary and tertiary industries | 0.0424 | 0.0847 | + |
| | Economic urbanization | GDP growth rate | 0.0531 | 0.0853 | + |
| | | Disposable income of urban and rural residents | 0.0725 | 0.0908 | + |
| | | The proportion of output value of secondary and tertiary industries | 0.0394 | 0.0858 | + |
| | | The proportion of fiscal expenditure on science and education in GDP | 0.0903 | 0.0867 | + |
| | Social urbanization | Per capita retail sales of consumer goods | 0.0712 | 0.0910 | + |
| | | Urban bus operation volume | 0.2174 | 0.1091 | + |
| | | PM2.5 | 0.0897 | 0.0874 | − |
| | Spatial urbanization | Per capita built-up area | 0.0985 | 0.0906 | + |
| | | The proportion of urban built-up area in land area | 0.1791 | 0.1015 | + |
| LE | Population pressure | Population density | 0.0222 | 0.0674 | − |
| | Industry pressure | Total agricultural output value-added/The area of cultivated land acquired in the year | 0.0215 | 0.0670 | − |
| | Environmental pressure | Fertilizer application per unit sowing area | 0.0526 | 0.0685 | − |
| | | Industrial waste gas, wastewater, and waste load per unit of land | 0.0331 | 0.0678 | − |
| | | The proportion of industrial land area | 0.0188 | 0.0672 | − |
| | Land structure | Urban per capita green area | 0.1715 | 0.0771 | + |
| | | Total water resources/Crop sown area | 0.2163 | 0.0860 | + |
| | Land function | Grain output per unit sown area | 0.0659 | 0.0694 | + |
| | | land economic density | 0.1679 | 0.0842 | + |
| | Overall development | Green development index | 0.0785 | 0.0721 | + |
| | Population response | population growth rate | 0.0153 | 0.0668 | − |
| | Industry response | GDP per capita | 0.0839 | 0.0725 | + |
| | Environmental response | The comprehensive utilization rate of general industrial solid waste | 0.0270 | 0.0671 | + |
| | | Green coverage rate | 0.0256 | 0.0670 | + |

### 3.2.2. CRITIC + Entropy Weight Method

Referring to the current reasonable practice of linearly combining the weight coefficients calculated by different weighting methods. The CRITIC method and the entropy weight method are applied to comprehensively determine the weight coefficients of each indicator and calculate the corresponding system development index. Where the CRITIC method takes into account the contrast strength and conflict between the indicator data. The entropy weight method emphasizes the dispersion between the indicator, which can make up for the deficiency of the CRITIC method in this aspect. Given this, the CRITIC method and the entropy weight method are used to construct a comprehensive weighting model in this paper, and the specific calculation process is as follows:

Step 1: the following model can be constructed according to the formula of the CRITIC method:

$$c_j = \frac{\sigma_j}{\overline{x}_j} \sum_{l=1}^{m} \left(1 - \left|r_{lj}\right|\right) \tag{1}$$

$$W_j^{(1)} = \frac{c_j}{\sum\limits_{j=1}^{m} c_j} \tag{2}$$

where $m$ represents the total number of indicators. $\sigma_j$ and $\overline{x}_j$ are the standard deviation and mean values of the standardized indicators of $j$. $r_{lj}$ is the correlation coefficient of the indicators of $l$ and $j$. $c_j$ indicates the information quantity of the indicator $j$. $W_j^{(1)}$ is the CRITIC weight of the indicator of $j$.

Step 2: Calculate the weights based on the entropy weight method. Since the ordinary entropy weight method has limitations in the longitudinal comparison in time, the time dimension is introduced with the following formula:

$$D_{tij} = x_{tij} / \sum_{t=1}^{q} \sum_{i=1}^{n} x_{tij} \tag{3}$$

$$e_j = -\frac{1}{\ln qn} * \sum_{t=1}^{q} \sum_{i=1}^{n} D_{tij} \ln D_{tij} \tag{4}$$

$$W_j^{(2)} = 1 - e_j / \sum_{j=1}^{m} 1 - e_j \tag{5}$$

where $q$ indicates the length in years. $n$ indicates the number of cities. $D_{tij}$ is the indicator proportion of the $j$ indicator of city $i$ for year $t$. $x_{tij}$ is the standardized indicator data. $e_j$ is the entropy value of the $j$ indicator. $W_j^{(2)}$ is the weight of the $j$ indicator, which is calculated by the entropy weight method.

Step 3: Combine the weights and calculate the subsystem composite development index as follows:

$$W_j = \alpha W_j^{(1)} + (1 - \alpha) W_j^{(2)} \tag{6}$$

$$S_{ti} = \sum_{j=1}^{m} \left(W_j * x_{tij}\right) \tag{7}$$

where $W_j$ is the composite weight of the $j$ indicator; this paper considers that the two weighting methods have the same importance, so $\alpha = 0.5$. $S_{ti}$ is the composite development index of city $i$ for year $t$.

### 3.2.3. Modified Coupled Coordination Model

The concept of coupling first came from electromagnetism, used to indicate the physical phenomenon that different electromagnetic waves affect each other and tend to be synergistic. Later the meaning was extended and widely used in economic, social, urban, industrial, and other fields. Currently, the coupled coordination model has become an effective research tool for evaluating the balanced development of social, economic, ecological, and industrial systems. In terms of urban development evaluation, the coupled coordination model can objectively represent the interrelationships and synergistic trends among different systems within a city. Based on this, the coupled coordination model is developed to analyze the multiple coupling relationships and the coordination level among DE, NU, and LE in the Yangtze River Delta, where the coupling degree is a measure of the association degree among subsystems, reflecting the strength of the interaction degree. The effect is not divided into pros and cons. Coordination refers to the benign interaction relationship among subsystems. Therefore, the coordination degree can measure the degree

of mutual synergy among subsystems and reflect the process of the overall system from disorder to order. However, the traditional coupling coordination model has problems such as over-reliance on the development of the system itself, weak validity of model use, and uneven distribution of coupling degree. Therefore, referencing the existing related studies [65], this paper further constructs a modified ternary coupling coordination model as follows:

$$C = \sqrt{\left[1 - \frac{\sqrt{(U_3 - U_2)^2 + (U_3 - U_1)^2 + (U_2 - U_1)^2}}{3}\right] * \sqrt{\frac{U_2 U_1}{U_3^2}}} \tag{8}$$

$$T = \delta_1 U_1 + \delta_2 U_2 + \delta_3 U_3, \ \delta_1 + \delta_2 + \delta_3 = 1 \tag{9}$$

$$D = \sqrt{C * T} \tag{10}$$

where $C$ is the coupling degree among the subsystems and values range from [0, 1]. From its formula, it can be seen that the closer the development indexes of the subsystems, the greater the coupling degree. It indicates the stronger association among the subsystems. $U$ represents the DE, NU, and LE systems, both values range from [0, 1], where $U_3$ is the maximum value among the subsystems. $T$ is the comprehensive development index of the subsystems. $\delta$ is the undetermined coefficient for each system, and this paper takes its value as 1/3 because of the consistent importance of the subsystems. $D$ is the coupling coordination degree of the subsystems and the values range from [0, 1], the closer the value is to 1, the higher the level of coordinated development of the subsystems. According to the magnitude of the coupling coordination degree of the subsystems and the research needs, the coupling coordination type of each city in the Yangtze River Delta is divided into three types and refined into five stages, as shown in Table 2.

**Table 2.** Type division of coupling coordinated development.

| Development Type | Coupling Coordination Degree | Coupling Coordination Stage |
|---|---|---|
| Coordinated development | $0.7 < D \leq 1$ | Good coordination |
| | $0.6 < D \leq 0.7$ | Intermediate coordination |
| Transformation development | $0.5 < D \leq 0.6$ | Basic coordination |
| | $0.4 < D \leq 0.5$ | Basic imbalance |
| Imbalance development | $0.3 < D \leq 0.4$ | Intermediate imbalance |
| | $0 < D \leq 0.3$ | Serious imbalance. |

### 3.2.4. Kernel Density Estimation

In socio-economic studies, kernel density estimation is often used to describe the distribution pattern of economic events and the evolution process of the data. Therefore, the overall distribution and evolutionary trajectory of the coupling coordination of the subsystems are described by the kernel density estimation method in the nonparametric estimation method. When the data obey the same distribution, the formula is as follows:

$$f(x) = \frac{1}{nh} \sum_{i=1}^{n} k\left(\frac{a_i - \overline{a}}{h}\right) \tag{11}$$

where $n$ is the number of observed objects. $h$ is the density estimation bandwidth. $a_i$ and $\overline{a}$ are the correlation index and its mean value, respectively. $k$ is the stochastic kernel function, and here a Gaussian kernel function is used.

### 3.2.5. PVAR Model

There are complex relationships among DE, NU, and LE that are intertwined and influence each other. Thus, there may be endogeneity problems in using ordinary models to identify their coupling mechanism. Based on this, the establishment of a PVAR model

can better overcome these problems and also identify the interaction of the subsystems, the specific form is as follows:

$$Y_{it} = \beta_0 + \sum_{f=1}^{p} \beta_f Y_{i,t-f} + \mu_i + \nu_t + \varepsilon_{it} \tag{12}$$

where $Y_{it}$ is the matrix of explained variables in this model, including the log values of DE, NU, and LE. $\sum_{f=1}^{p} \beta_f$ is the matrix of estimated coefficients. $\beta_0$ is the constant term. $p$ is the lagged order. $\mu_i$ reflects city-fixed effects. $\nu_t$ reflects year-fixed effects. $\varepsilon_{it}$ is the random error term.

3.2.6. Spatial Panel Model

Cities, as typical spatial units, are bound to have some degree of spatial association, and the spatial econometric model is an effective tool to study the spatial effects. Based on this, through the test of spatial correlation and the comparison of different spatial econometric models, according to the study purpose and theoretical analysis, this paper sets up a spatial Durbin panel model of the following form:

$$D_{it} = \eta_0 + \rho W * D_{it} + \sum_{d=1}^{3} \eta_d U_{dit} + W * \sum_{d=1}^{3} \gamma_d U_{dit} + \phi Z_{it} + \theta W * Z_{it} + \nu_i + \mu_t + \varepsilon_{it} \tag{13}$$

where $D_{it}$ denotes the coupling coordination degree. $U$ denotes the DE, NU, and LE. $Z_{it}$ is the control variables. $W$ denotes the spatial weight matrix normalized by rows. $\rho$ denotes the spatial autoregressive coefficient, which is used to measure the influence of coupling coordination degree of neighboring cities on local cities. $\gamma_d$ is the spatial lagged terms of the DE, NU, and LE, which is used to measure the neighbor effects of the subsystems. $\theta$ is the spatial lagged terms of the control variables. The rest of the variables are consistent with the above.

## 4. Empirical Analysis

*4.1. Time Evolution Analysis of the Overall Coupling Coordination Development of the Yangtze River Delta*

Based on the modified coupling coordination model, the development indexes and coupling coordination degree of the DE, NU, and LE in the Yangtze River Delta are measured for each year from 2011 to 2020. The results are shown in Table 3 and Figure 2.

**Table 3.** Development indexes and coupling coordination degree of DE–NU–LE in the Yangtze River Delta.

| Year | DE | NU | LE | Coupling Coordination Degree | Coupling Degree | Coupling Coordination Stage |
|------|--------|--------|--------|------|------|------|
| 2011 | 0.0943 | 0.3082 | 0.4152 | 0.3811 | 0.5350 | Intermediate imbalance |
| 2012 | 0.1213 | 0.3315 | 0.4290 | 0.4120 | 0.5802 | Basic imbalance |
| 2013 | 0.1514 | 0.3180 | 0.4271 | 0.4303 | 0.6222 | Basic imbalance |
| 2014 | 0.1647 | 0.3207 | 0.4359 | 0.4388 | 0.6298 | Basic imbalance |
| 2015 | 0.1799 | 0.3359 | 0.4446 | 0.4545 | 0.6487 | Basic imbalance |
| 2016 | 0.2037 | 0.3567 | 0.4498 | 0.4776 | 0.6814 | Basic imbalance |
| 2017 | 0.2345 | 0.3683 | 0.4463 | 0.5000 | 0.7193 | Basic coordination |
| 2018 | 0.2546 | 0.3837 | 0.4543 | 0.5159 | 0.7360 | Basic coordination |
| 2019 | 0.2690 | 0.3981 | 0.4689 | 0.5278 | 0.7418 | Basic coordination |
| 2020 | 0.2811 | 0.4011 | 0.4724 | 0.5353 | 0.7510 | Basic coordination |

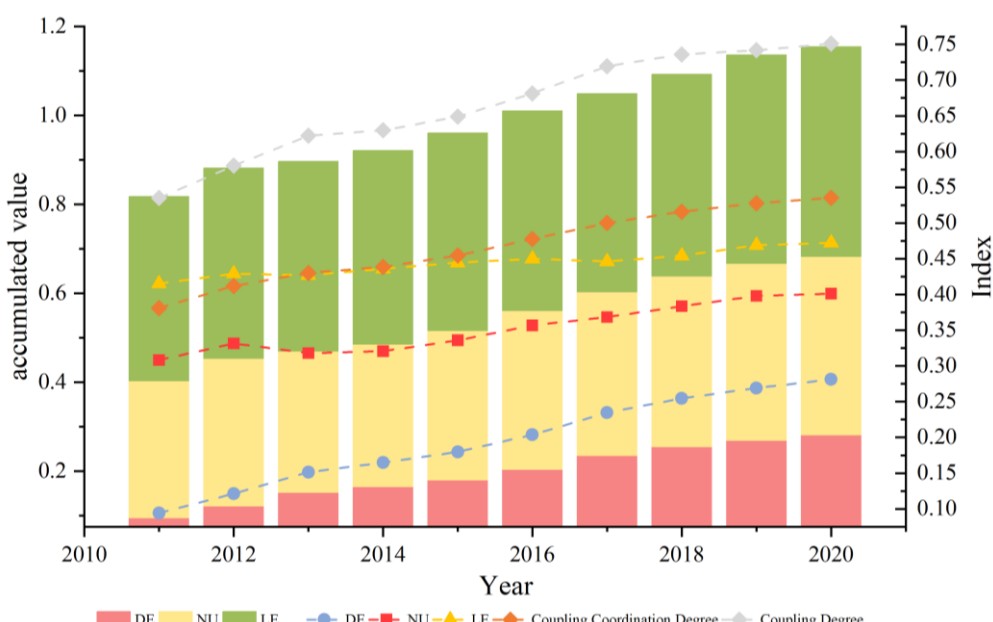

**Figure 2.** DE–NU–LE coordinated development trends in the Yangtze River Delta.

According to Table 3 and Figure 2, in the Yangtze River Delta, the overall coupled and coordinated developments of the DE, NU, and LE show a stable upward trend from 2011 to 2020. The coupling degree is always higher than the coupling coordination degree and subsystems development indexes. The coupling value increases from 0.54 in 2011 to 0.71 in 2020. The result indicates that the interaction among the subsystems of DE, NU, and LE in the past 10 years has been continuously deepened, and tends towards positive development. The coupling coordination degree increases from 0.38 in 2011 to 0.54 in 2020, which achieves the transition development from moderate disorders to basic coordination as a whole. In terms of the development of subsystems, the development level of DE, NU, and LE are significantly different from 2011 to 2020 but has a gradually converging trend over time. Among them, the LE system shows a slow and fluctuating growth with a brief decline in 2013 and 2017. It has always been in the leading position and has become an important supporting part of the ecological environment of the Yangtze River Delta. The developments of NU and DE show a steady upward trend in general. It is worth noting that NU shows a short decline and stagnation from 2012 to 2014, and then shows a more moderate growth trend. The reason may be that the Yangtze River Delta region, as a highland of policy innovation and reform, had already started relevant constructions before the official launch of NU in 2014. After a short throe of deep reforms, it has gradually promoted the urbanization transition from traditional to new-type. The high-quality development of multi-dimension has become the goal of each city. Similarly, after three years of rapid growth, the growth rate of DE slowed down after 2013. Under the joint action of fluctuation growth of the LE, urbanization reforms, and the slowdown of the DE development, the growth of the coupling coordination degree has also experienced a process of "slowing down-accelerating-slowing down again" after 2012.

*4.2. Spatial and Temporal Evolution Characteristics of DE–NU–LE and Coupling Coordination Degree in the Yangtze River Delta Prefecture-Level Cities*

4.2.1. The Foundations for Coordinated Development of Each City Have Begun to Form, and the Trend of DE Polarization Is Obvious

Based on the indicators in Table 1 above, the development indexes of DE–NU–LE of each prefecture-level city in the Yangtze River Delta are calculated by the CRITIC + entropy weight method. Figure 3 demonstrates the changes of DE–NU–LE and the coupling coordination degree over different years.

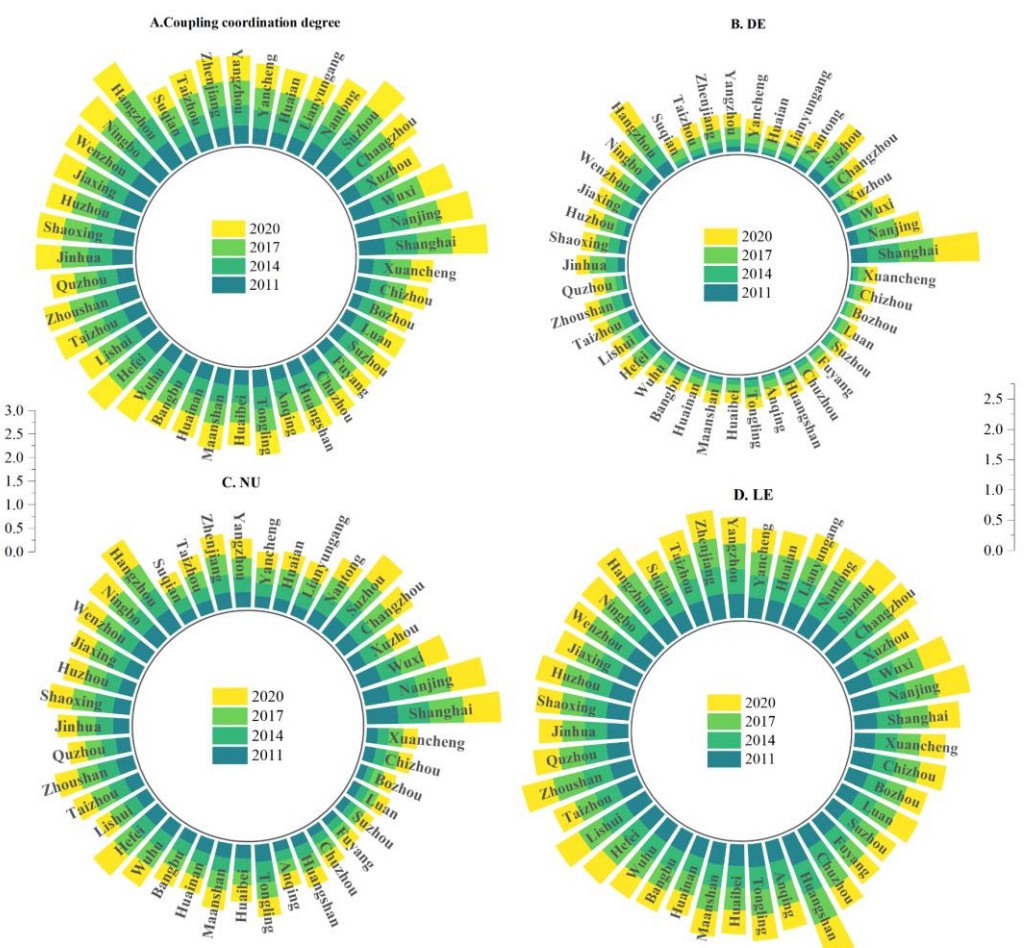

**Figure 3.** The changes of DE–NU–LE and the coupling coordination degree of each prefecture-level city in the Yangtze River Delta over different years. ((**A**) presents the change of the coupling coordination degree. (**B**) presents the change of DE. (**C**) presents the change of NU. (**D**) presents the change of LE).

Figure 3A represents the changes over different years in the coupling coordination degree of each prefecture-level city in the Yangtze River Delta. It can be seen that the coupling coordination degree of each city has developed significantly in each period, showing a clear gradient pattern with a distinct circle structure. Shanghai and Hangzhou are at the first level. In particular, Shanghai, as the only municipality directly under the central government in the Yangtze River Delta, has always been in the leading position regarding coupling coordination degree. Nanjing, Wuxi, Suzhou, and Ningbo are at the second level with good geographical locations and development foundations. The rest of the cities are located at the third level, with a relatively low coupling coordination degree. It is worth noting that from 2014 to 2017, the coupling coordination degree of the cities had the most significant increase, with the overall average level rising from 0.44 to 0.50. During this period, the connotation of NU is greatly enriched, and the ecological civilization construction and the idea of sustainable development gradually gained popularity. Meanwhile, with the arrival of the mobile digital era, the related digital constructions such as digital city and smart city continuously strengthen the coupling state among DE, NU, and LE. Further, the benign interaction can be promoted to complete the stage transformation of the overall coordinated development from imbalance to coordinated.

In terms of the changes of the subsystems, the development of DE shows a pattern with the three pillars of Shanghai, Nanjing, and Hangzhou. The low value is in Anhui Province, which is basically lower than the regional average (Figure 3B). The development

of the DE requires sustained talent investment and technological breakthroughs. Shanghai, Hangzhou, and Nanjing not only have strong economic strength, but also have abundant talent reserves and complete basic information facilities. As regional central cities, they have certain advantages in resource utilization efficiency and policy tendency. From the perspective of local administrators, the leading DE development of Shanghai, Hangzhou, and Nanjing cannot be achieved without the attention of local governments. By combing through the local plans of DE development, it is easy to find that Hangzhou and Shanghai managers are more focused on enhancing the development drivers of the DE and the expansion of related application fields. They are committed to establishing themselves as the leading highland of DE development by breakthroughs in new technologies and new fields. Nanjing is more focused on the integration of the DE with production, life, and ecology, to enhance its universality and inclusion. While Anhui Province is dominated by agriculture and traditional industries, with a weak foundation for digital development, the digital industry has not yet formed a scale. Hence, the digital gap with other cities is becoming increasingly obvious. The NU shows a relatively balanced development (Figure 3C). Among them, Shanghai, Nanjing, Suzhou, Hangzhou, and Hefei have great guiding functions, but the NU level of some cities in Anhui Province and north of Jiangsu Province still needs to be improved. The circle shape of LE is relatively plump and presents an overall balanced development during the study period (Figure 3D). The result points out that all cities in the Yangtze River Delta have a good foundation of LE. The periodic increase is small, indicating that the improvement of LE has a certain pressure. The superior land foundation benefits from the natural geographical advantages of the Yangtze River Delta. First of all, the Yangtze River Delta has good agricultural foundations with a dense river network and fertile soil. Secondly, the inland cities are mainly hilly areas with stable geological conditions, abundant water sources, and low incidence of natural disasters. Correspondingly, the solid land environment also means that the improvement of LE is not easy.

### 4.2.2. The Spatial Evolution Characteristics of Subsystems Are Different, and the Spatial Pattern of "High in the East and Low in the West" Is Gradually Strengthened

Columns A, B, and C of Figure 4 show the spatial distribution characteristics of DE, NU, and LE in the Yangtze River Delta from 2011 to 2020, respectively. First of all, from the spatial evolution characteristics of DE, Shanghai has the highest score of 0.33 in 2011, followed by Hangzhou at 0.25. The overall DE development level is low, and most of the cities are in "digital winter", especially in the northern part of Jiangsu Province and the whole area of Anhui Province, which shows the characteristics of low-level clustering of DE. In 2014, DE development shows a trend of gradually spreading outward with Shanghai, Hangzhou, and Nanjing as the center. In 2020, Shanghai, Hangzhou, and Nanjing had become important advantageous cities for DE development in the Yangtze River Delta, and the development gap with surrounding cities continues to widen. It is visible that in the Yangtze River Delta, the DE is still in the early stage of development and requires a large amount of resource investment. Shanghai, Nanjing, and Hangzhou have gradually become the digital growth poles by their scale advantages and digital foundation. In general, the development of the DE in the Yangtze River Delta is significantly polarized, and the overall development level still has a lot of room for improvement.

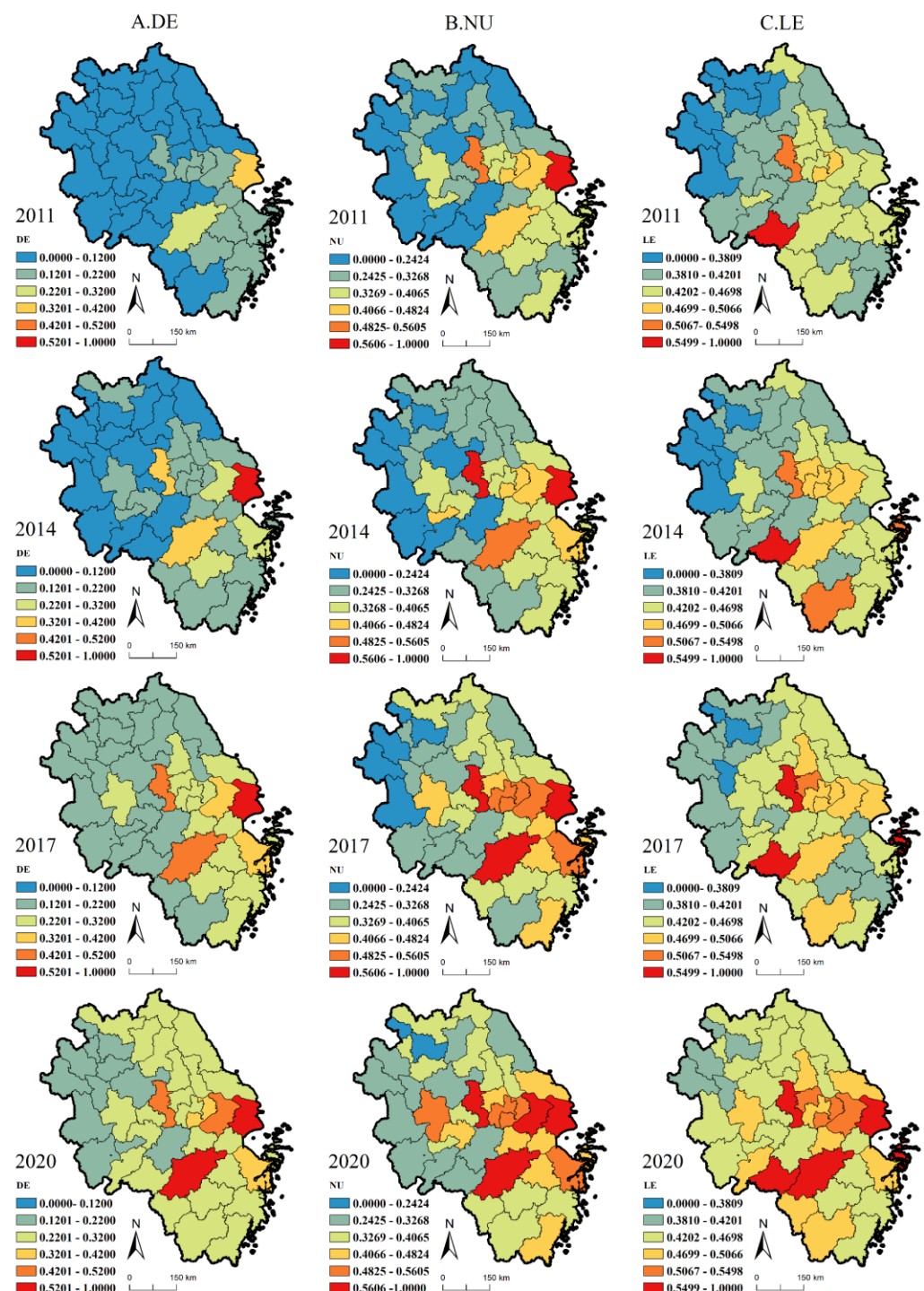

**Figure 4.** Spatial evolution characteristics of DE–NU–LE. ((**A**) presents the four-year changes of DE. (**B**) presents the four-year changes of NU. (**C**) presents the four-year changes of LE).

Secondly, from the spatial evolution characteristics of NU, Shanghai still achieved the highest score at 0.64 in 2011, followed by Nanjing and Hangzhou at 0.52 and 0.46, respectively. Some low values still clustered in the northern and western part of the Yangtze River Delta. Compared with the polarized advantage of DE in individual cities, NU shows a more balanced development. The Z-shaped distribution with metropolitan areas of Shanghai, Nanjing, Hefei, Hangzhou, and Ningbo as the main axis has gradually formed in 2014 to 2020. The Z-shaped pattern of NU is highly consistent with the pattern of population distribution and economic development in the Yangtze River Delta. This

phenomenon indicates that population and economy are the basis for the sustainable promotion of NU. The NU with the people-oriented concept cannot be separated from individual labor and development, and the economic foundation is the basic guarantee for the continuous deepening of NU. On the whole, the overall development of NU in the Yangtze River Delta shows a positive trend from 2011 to 2020, and the gap between cities has gradually narrowed. The spatial distribution presents an obvious Z-shaped pattern, and continuously deepens into the central, southern, and northern regions.

Finally, LE exhibits a very different spatial distribution from that of the DE and NU. Huangshan has the highest score of 0.57 in 2011, followed by Nanjing and Wuxi at 0.51 and 0.47, respectively. Most cities obtain scores between 0.3 and 0.45 with small differences between cities, and a small number of low values gather in the northwest of the Yangtze River Delta. In 2014–2020, the spatial distribution pattern with "Chizhou-Lishui" and "Nanjing-Zhoushan" as two horizontal lines, and "Quzhou-Nantong" as one vertical line has gradually formed. Compared with the spatial distribution patterns of DE and NU, the spatial evolution path of LE shows obvious ecological characteristics. Although the economic development and urbanization construction of Huangshan, Quzhou, Lishui, Chizhou, and Zhoushan are relatively backward. They still have significant ecological advantages and green development potential with abundant natural resources, high vegetation coverage, and stable ecological environment, forming an important ecological protective screen in the Yangtze River Delta. Besides, in these cities, local governments strongly support eco-industries such as tourism, wellness, and culture to promote them as a new green driving force for local development. Nanjing, Suzhou, Shanghai, Hangzhou, and Wuxi not only take the lead in economic development and urban construction, but also have good performance in LE. This shows that the good development of LE not only comes from the natural advantages, but also the active practice of the ecological concept by governments, which feeds the achievements of economic development back into the ecological construction. Generally, the development of LE in the Yangtze River Delta tends to be balanced from 2011 to 2020, and the development pattern of two horizontal lines and one vertical line has emerged.

The spatial distribution characteristics of the coupling coordination degree of the subsystems in the Yangtze River Delta from 2011 to 2020 are shown in Figure 5. In terms of the coupling coordination degree, the spatial distribution of coupling coordination is dominated by low-level and non-uniform distribution in the early stage. The reason may be that the early DE and NU are at a low-level of development, and it is difficult to form a good interactive relationship with LE. In the middle and late period of the study, the development gap among DE, NU, and LE gradually narrowed, and even overtook in some cities. The coupling degree of the subsystems is increasing, the coupling coordination degree is distributed in sheets. The pattern of "high in the east and low in the west" is more and more significant.

From the perspective of the spatial evolution path, Shanghai, Hangzhou, Nanjing, and Suzhou, only four cities, are in the basic coordination stage in 2011, while the rest are in the imbalance stage; and where Anqing, Chuzhou, Lu'an, Suzhou, Fuyang, and Bozhou are in the serious imbalance stage. At this time, only Shanghai and Nanjing have advanced development of NU, while the rest of the cities have certain advantages in LE. In 2014, Shanghai, Hangzhou, and Nanjing entered the intermediate coordination stage. Wuxi, Ningbo, Jinhua, and Wenzhou moved from the basic imbalance stage to the basic coordination stage, and Bozhou is still in the serious imbalance stage. Hangzhou changed from the advanced development of LE to the advanced development of NU. In 2017, Shanghai took the lead in entering the good coordination stage, and Suzhou moved from the basic coordination stage to the intermediate coordination stage. With the approval of the Yangtze River Delta urban agglomeration in 2016, the integrated development of the Yangtze River Delta has become a national proposition, and the connection of cities has been further strengthened. However, only partial cities in Anhui Province have joined the Yangtze River Delta urban agglomeration. Thus, the map shows that the large number of

yellow patches deepen to the southern region, and green patches spread to the northern region. Meanwhile, Nanjing, Wuxi, Suzhou, Shanghai, Jiaxing, Hangzhou, Shaoxing, Ningbo, and Hefei have advanced development of NU, which indicates that NU has certain development advantages in regional central and sub-central cities. The leading cities of NU are mainly distributed on the Z-shaped axis, which is consistent with the evolution characteristics of NU above. *The outline of the integrated regional development of the Yangtze River Delta* in 2019 includes the whole region of Anhui Province in the development plan of the Yangtze River Delta. The coordinated development of the Yangtze River Delta is pushed towards a new stage. Therefore, in 2020, more than half of cities in the Yangtze River Delta enter the coordination stage, and Hangzhou shifts from the intermediate coordination stage to the good coordination stage. The yellow patches begin to cover the green patches from east to west and south to north. Hefei, Wuhu, and Maanshan, which are close to Nanjing and Hangzhou, have taken the lead in developing well in Anhui Province due to the effect of proximity. The rest of the cities in Anhui Province are still relatively backward in coordinated development because of the weak development foundation, even though they are radiated by central cities. The gradation pattern of "high in the east and low in the west" becomes clearer. At this time, Shanghai has greatly advanced development of DE, and the development advantage of NU is still concentrated in some central cities that are on the Z-shape axis. The remaining cities are still dominated by the advantages of LE. The advance of LE and the relative lag of DE and NU in most cities may cause changes in the internal functions and structures of the city system. The coordinated development of the subsystems may be restricted to a certain extent.

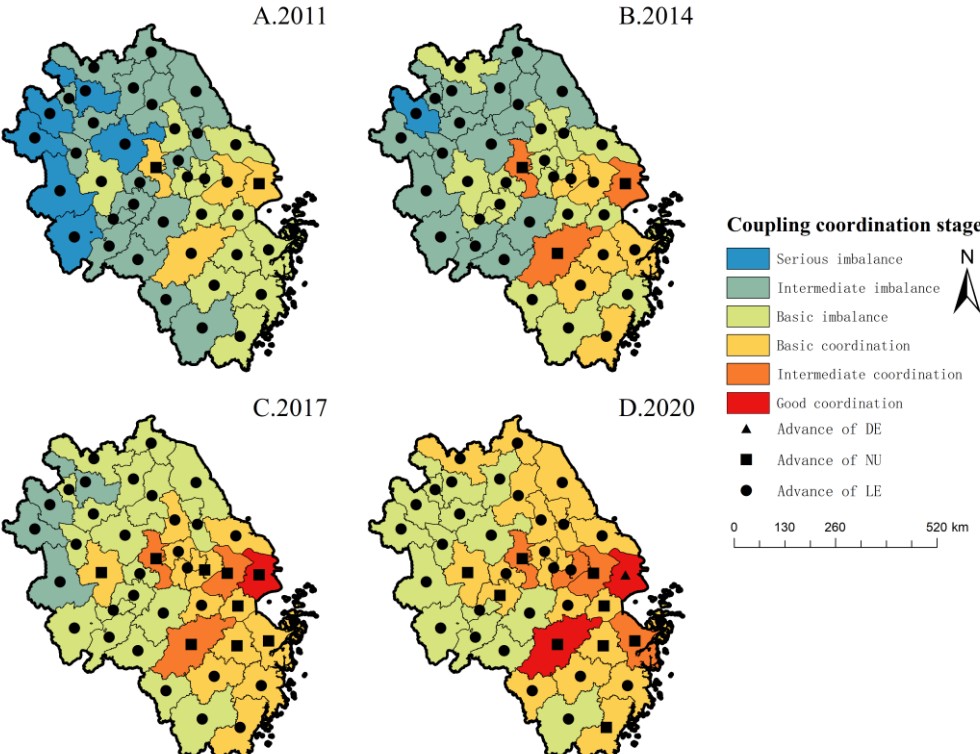

**Figure 5.** Spatial evolution characteristics of the coupling coordination degree. ((**A**) presents the change of the coupling coordination degree in 2011. (**B**) presents the change of the coupling coordination degree in 2014. (**C**) presents the change of the coupling coordination degree in 2017. (**D**) presents the change of the coupling coordination degree in 2020).

### 4.2.3. The Tailing of the Right Side Is Apparent, and Regional Development Gradually Converges

From the above analysis, it can be seen that the overall coordinated development of DE, NU, and LE in the Yangtze River Delta has stably improved, and each city initially has the foundation of coordinated development, while the evolutionary trajectory of the subsystems varies and development differences still exist. Therefore, kernel density estimation is further used to reveal the dynamic evolution characteristics of the coupling coordination degree and the subsystems (Figure 6).

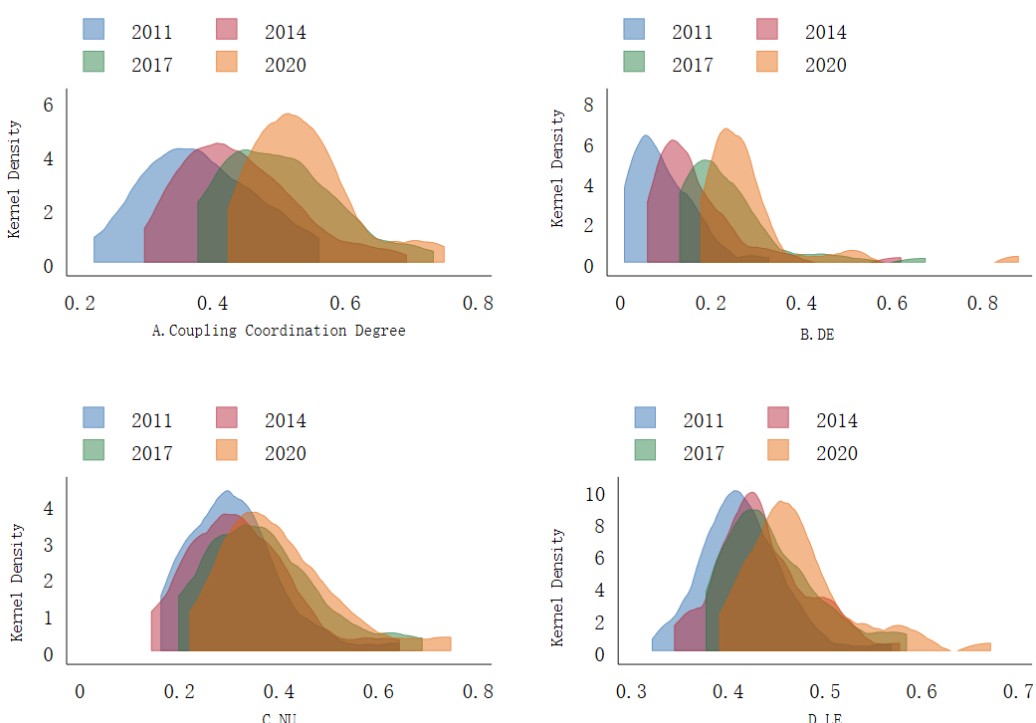

**Figure 6.** Kernel density analysis of the coupling coordination degree and the subsystems. ((**A**) represents the kernel density change of coupling coordination degree. (**B**) represents the kernel density change of DE. (**C**) represents the kernel density change of NU. (**D**) represents the kernel density change of LE).

From the position covered, the overall kernel density curve of coupling coordination degree gradually shifts to the right. This moving trend indicates that the overall coordinated development level of the subsystems in the Yangtze River Delta is gradually improving. In 2011–2014 and 2014–2017, the span of the curve shifting to the right is more obvious than other periods, which indicates that the growth rate of coordinated development level has accelerated during this period. From the shape of the curve, the kernel density curve shows a more obvious trailing feature to the right from 2014 to 2020. The wave width tends to contract and the wave peak gradually becomes larger. These characteristics indicate that the gap between the coordinated development levels of the Yangtze River Delta cities are gradually decreasing. However, the phenomenon of individual cities that have unique development advantages such as Shanghai still exist. In terms of the evolution of the subsystems, the kernel density curve of DE shows an obvious trend of moving to the right, with an overall slender shape, and the trailing feature to the right intensifies over time. All of these characteristics suggest two phenomena. One is that the development gap among the cities in the Yangtze River Delta tends to narrow. The other is that individual cities such as Shanghai, Nanjing, and Hangzhou show certain accumulative advantages and are far ahead in the development of DE. The kernel density curves of NU and LE both show a similar overlapping trend during the study period, which indicates a slower pace of development for NU and LE. Additionally, both the tailings of the right side tend to become

thicker and longer, especially LE, with a trend of change from unipolar to multipolar development, indicating that the development of LE is reflected at all levels, and the overall structure tends to be stable.

## 5. Identification of the Interaction Mechanisms and Spatial Effects of DE, NU, and LE

*5.1. Interactive Response Identification of DE, NU, and LE*

To further explore the dynamic interaction effects of the DE, NU, and LE, this paper employs the PVAR model to quantitatively analyze the dynamic relationships among the subsystems. To avoid the pseudo-regression problem caused by the root unit, the original data are logarithmically processed to make them stationary. The results of LLC and IPS tests are shown in Table 4.

**Table 4.** Data stationarity test.

| Variance | LLC | IPS | Individual Fixed | Time Trend | Test Results |
|---|---|---|---|---|---|
| ln_DE | −34.5572 *** | −11.4447 *** | Yes | Yes | Stationary |
| ln_NU | −8.8700 *** | −3.3008 *** | Yes | Yes | Stationary |
| ln_LE | −16.5960 *** | −5.2458 *** | Yes | Yes | Stationary |

* Note: *** indicate significance at the 1% statistical levels.

The Grange causality test in Table 5 indicates that there is a two-way causal relationship between DE, NU, and LE. The optimal lag period is determined to be two according to the AIC criterion, so the PVAR (2) model is established. Based on this, the impulse response analysis of DE, NU, and LE is carried out with Stata to explore the dynamic interaction and response trends among the subsystems. The results are shown in Figure 7.

**Table 5.** Granger causality test.

| Equation | Excluded | *p*-Value | Whether to Reject the Null Hypothesis |
|---|---|---|---|
| ln_DE | ln_NU | 0.002 *** | Rejection |
| | ln_LE | 0.048 ** | Rejection |
| | All | 0.000 *** | Rejection |
| ln_NU | ln_DE | 0.000 *** | Rejection |
| | ln_LE | 0.060 * | Rejection |
| | All | 0.000 *** | Rejection |
| ln_LE | ln_DE | 0.018 ** | Rejection |
| | ln_NU | 0.005 *** | Rejection |
| | All | 0.001 *** | Rejection |

* Note: *, **, and *** indicate significance at the 10%, 5%, and 1% statistical levels, respectively. The null hypothesis: B is not the Granger cause of A.

The following results can be found in Figure 7: (I) The responses of DE, NE, and LE to their own standard deviation information shocks in the Yangtze River Delta show a significant positive response (Figure 7A,E,I). The responses are the largest at the beginning of the period and then begin to weaken until it tends to be stable. This trend indicates that the development of each subsystem in the Yangtze River Delta has positive progressive effects and inertial development characteristics. Where the self-reinforcing effect of DE is the most persistent until the 10th period. The self-reinforcing effect of NU is relatively short-lived until the 5th period. The self-reinforcing effect of LE continues until the 9th period. The above descriptions indicate that the development of each subsystem in the Yangtze River Delta shows different degrees of self-reinforcing effects and path-dependent characteristics, and both weaken over time. (II) When facing the shock of NU, DE shows a significant positive response at the beginning of the period (Figure 7B) and reaches a peak in period 1. Then it starts to decline and turn into a negative response and tends to stabilize from period 4, with insignificant response. Similarly, DE does not respond significantly to the shock of LE (Figure 7C). With the further deepening of NU and the extension of

ecological concepts, the digital technology and its application have received unprecedented attention. However, the current urbanization quality and ecological civilization construction still need to be improved, which means that the stimulation for the development of DE is limited. Especially, the support for related technology is weak and has not yet formed a breakthrough power to promote it. (III) NU shows a significant positive response in general when facing the shock of DE. It peaks in period 3 after a short period of ups and downs, and then starts to fall back and stabilizes after period 14 (Figure 7D). This indicates that DE shows sustaining growth momentum and development potential in the process of integration with NU, which improves the level of urban digitization. For the shock of LE, NU shows a non-significant negative response (Figure 7F). Although LE plays a supportive role for NU, the land contradiction between urban development and land conservation has been long-standing, especially in the early crude city development mode, which has problems such as weak ecological awareness and over-exploitation of land. It is difficult to support the NU through the function of resource guarantee and wealth appreciation. (IV) LE shows a significant positive response in general when faced with the shock of DE (Figure 7G). The initial response is the strongest, then weakens in a fluctuating manner and tends to be stable. This indicates that with the DE development, the application of digital technology related to digital planning, intelligent land monitoring, and digital governance has effectively improved the city's ability to protect LE. For the shock of NU, LE shows a significant positive response at the beginning of the period and becomes negative and insignificant in period 1, and then rebounds rapidly (Figure 7H). This result indicates that the NU, which is oriented towards greenness and intensification, plays a certain guiding role in the protection of LE. However, the symbiosis between development and conservation still restricts the coordinated development of both, and even creates a stress effect between the two.

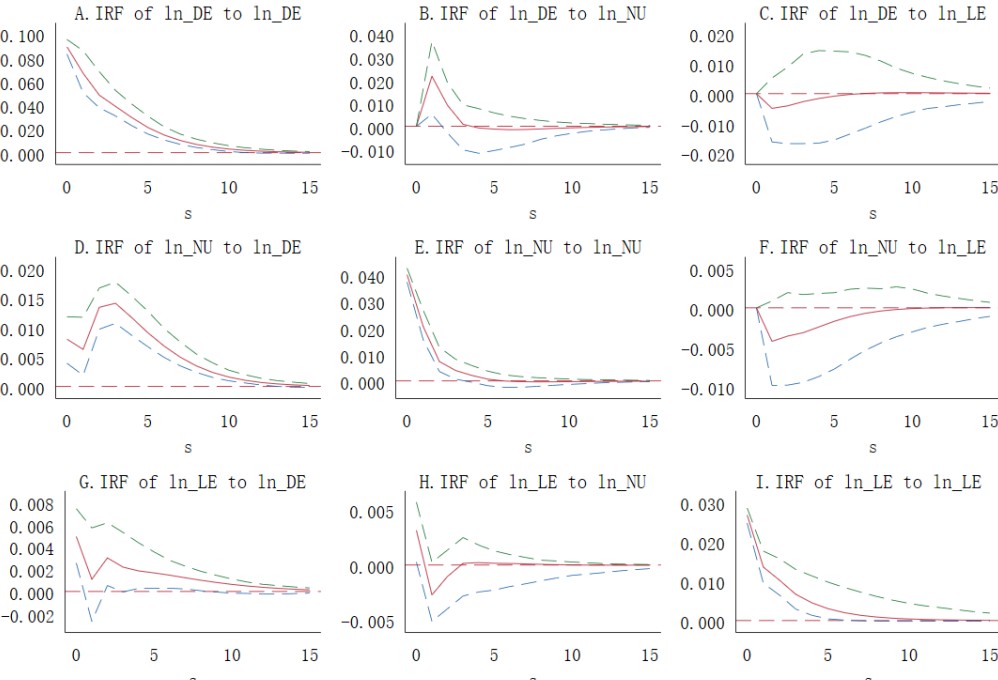

**Figure 7.** The impulse response analysis of DE, NU, and LE. ((**A**) represents the impulse response of ln_DE to ln_DE. (**B**) represents the impulse response of ln_DE to ln_NU. (**C**) represents the impulse response of ln_DE to ln_LE. (**D**) represents the impulse response of ln_NU to ln_DE. (**E**) represents the impulse response of ln_NU to ln_NU. (**F**) represents the impulse response of ln_NU to ln_LE. (**G**) represents the impulse response of ln_LE to ln_DE. (**H**) represents the impulse response of ln_LE to ln_NU. (**I**) represents the impulse response of ln_LE to ln_LE).

### 5.2. Spatial Effects Identification of DE–NU–LE Coordinated Development

The spatial and temporal differentiation characteristics and dynamic interaction mechanism of the DE, NU, and LE in the Yangtze River Delta from 2011 to 2020 are analyzed above. In order to further identify the spatial effects of the subsystems and the coupling coordination degree, a spatial panel model is established according to formula (13) based on verifying the spatial positive correlation of the coupling coordination degrees. Table 6 reports the test results of the spatial correlation of the coupling coordination degree in the Yangtze River Delta.

**Table 6.** Data stationarity test.

| Year | Adjacency Matrix | | | Economic Distance Matrix | | |
|---|---|---|---|---|---|---|
| | Moran's I | Z-Value | *p*-Value | Moran's I | Z-Value | *p*-Value |
| 2011 | 0.449 | 4.812 | 0.00 | 0.186 | 9.635 | 0.00 |
| 2012 | 0.460 | 4.918 | 0.00 | 0.187 | 9.665 | 0.00 |
| 2013 | 0.394 | 4.283 | 0.00 | 0.161 | 8.536 | 0.00 |
| 2014 | 0.394 | 4.314 | 0.00 | 0.154 | 8.268 | 0.00 |
| 2015 | 0.400 | 4.362 | 0.00 | 0.161 | 8.574 | 0.00 |
| 2016 | 0.386 | 4.211 | 0.00 | 0.157 | 8.376 | 0.00 |
| 2017 | 0.416 | 4.512 | 0.00 | 0.167 | 8.832 | 0.00 |
| 2018 | 0.400 | 4.512 | 0.00 | 0.162 | 8.605 | 0.00 |
| 2019 | 0.360 | 3.955 | 0.00 | 0.162 | 8.042 | 0.00 |
| 2020 | 0.318 | 3.548 | 0.00 | 0.130 | 7.200 | 0.00 |

From the results in Table 6, it can be found that the global Moran's I index values are all greater than 0 and significant at the 1% statistical level. It indicates that the spatial distribution of the coupling coordination degree in the Yangtze River Delta has a significant positive correlation. The prerequisite assumption of the spatial econometric model is satisfied. The global Moran's I index calculated by the different spatial weight matrices shows a fluctuating general downward trend during the study period. For the numerical value, the value of the global Moran's I index reaches its maximum value in 2012, when the spatial agglomeration of the coupling coordination degree is strongest. Additionally, it then begins to decline in a fluctuating manner. The index value in 2020 is smaller than its initial value, indicating that the spatial non-balanced development of the coupling coordination degree in the Yangtze River Delta has been improved after the continuous adjustment from agglomeration to dispersion.

In order to precisely identify their spatial effects and avoid estimation bias caused by omitted variables, the following variables are controlled: (I) The level of government income, measured by the proportion of government fiscal revenue in the GDP. (II) The level of government expenditure, measured by the proportion of government fiscal expenditure in the GDP. (III) The level of foreign investment, measured by the proportion of actually utilized foreign investment in the GDP. (IV) Consumption demand, measured by the proportion of retail sales of consumer goods in the GDP. Based on the above variables, a spatial panel model is constructed and the tests of corresponding spatial econometric model selection are conducted. The results in Table 7 show that both the LR test and the Wald test significantly reject the hypothesis that the spatial Durbin model (SDM) degenerates into the spatial autoregressive model (SAR) or the spatial error model (SEM) at the 1% statistical level. The Hausman test indicates that a fixed-effects model should be used.

**Table 7.** Selection of spatial econometric models.

| | SDM→SAR | | SDM→SEM |
|---|---|---|---|
| LR-test | 59.23 *** | | 42.47 *** |
| Wald-test | 64.56 *** | | 41.18 *** |
| Hausman | | 39.46 *** | |

* Note: *** indicate significance at the 1% statistical levels.



Table 8 reports the estimation results of the SDM based on different spatial weight matrices. The results show that regardless of which spatial weight matrix is used, the coefficients of DE and NU are significantly positive at the 1% statistical level. While the coefficient of LE is significantly negative at the 1% statistical level. Those results empirically prove the above conjecture that the development of DE and NU lags behind relative to LE in the Yangtze River Delta. This has to some extent caused uneven development among subsystems, which has a constraining effect on the overall coordinated development. The coefficient of $\rho$, which is the spatial lag term of the coupling coordination degree, is significantly positive. Indicating that the coordinated development degree of the subsystems has a significant spatial spillover effect, which can promote the coordinated development level of neighboring cities.

**Table 8.** Estimation results of the spatial Durbin model.

| Variance | (1) Adjacency Matrix | | (2) Economic Distance Matrix | |
|---|---|---|---|---|
| | ln_D | Wx | ln_D | Wx |
| ln_DE | 0.1872 *** | −0.0680 *** | 0.1873 *** | −0.0512 *** |
| | (0.0038) | (0.0126) | (0.0038) | (0.0088) |
| ln_NU | 0.2818 *** | −0.0250 | 0.2868 *** | −0.0335 ** |
| | (0.0105) | (0.0291) | (0.0103) | (0.0164) |
| ln_LE | −0.0611 *** | 0.0857 ** | −0.0478 *** | 0.0557 *** |
| | (0.0174) | (0.0344) | (0.0171) | (0.0215) |
| Control variables | Yes | | Yes | |
| $\rho$ | 0.2624 *** | | 0.2041 *** | |
| | (0.0660) | | (0.0440) | |
| N | 410 | | 410 | |
| R2 | 0.989 | | 0.988 | |

\* Note: Robust standard errors based on city level are in parentheses. \*\*, and \*\*\* indicate significance at the 5%, and 1% statistical levels, respectively. The following table is the same.

Since the regression coefficient of the spatial lag term is significantly not 0, simply using the regression coefficient to measure the spatial effect of each subsystem will have systematic bias. Hence, it is necessary to decompose the effect to obtain the direct and indirect effects of each subsystem, and the results are shown in Table 9.

**Table 9.** Decomposition of spatial effects.

| Variance | (1) Adjacency Matrix | | (2) Economic Distance Matrix | |
|---|---|---|---|---|
| | ln_D | | ln_D | |
| | Direct effect | Indirect effects | Direct effect | Indirect effects |
| ln_DE | 0.1862 *** | −0.0249 *** | 0.1865 *** | −0.0156 *** |
| | (0.0038) | (0.0065) | (0.0038) | (0.0037) |
| ln_NU | 0.2844 *** | 0.0653 ** | 0.2882 *** | 0.0305 ** |
| | (0.0104) | (0.0267) | (0.0102) | (0.0122) |
| ln_LE | −0.0551 *** | 0.0910 ** | −0.0427 *** | 0.0552 ** |
| | (0.0164) | (0.0428) | (0.0162) | (0.0244) |

\* Note: \*\*, and \*\*\* indicate significance at the 5%, and 1% statistical levels, respectively.

The results in Table 9 show that the direct effects of DE, NU, and LE are consistent with the above analysis results and show better robustness under different spatial weight matrices. This result indicates that the uneven development within the subsystems mainly restricts the coordinated development of local cities. In terms of indirect effects, the coefficient of DE is significantly negative, indicating that the DE development of local cities negatively affects the coordinated development of neighboring cities. This result is in line with the spatial and temporal evolution of DE. The reason behind it may be that the development of DE requires continuous breakthroughs in technology and continuous

investment of high-quality talents. As a result, regional diffusion cannot be formed in the short-term. Instead, it will produce a certain "siphon effect" on the resource endowment of neighboring cities, which is manifested in the accumulated advantages of digital development in individual cities such as Shanghai, Hangzhou, and Nanjing in reality. The indirect effects of NU and LE are both significantly positive, indicating that NU and LE have good spatial spillover effects. Particularly for the LE, the absolute value of the indirect effect is greater than the direct effect, which indicates that the improvement of regional coordination level is more dependent on the spatial spillover effect of LE. Additionally, it also reflects the publicity and universality of LE itself. Hence, accelerating the construction of NU and strengthening the protection of LE are not only beneficial to the development of local cities but also effectively promote the coordinated development of the region as a whole.

## 6. Conclusions and Suggestions

### 6.1. Main Conclusions

Based on the discussion of the coupling mechanism, the relative analysis described herein is applied to investigate the spatial and temporal characteristics of DE–NU–LE and the coupling coordination degree, with the Yangtze River Delta as the study area. Additionally, the dynamic interactions and spatial effects of LE, NU, and LE are further identified by the econometric model. The main findings are as follows:

First, in terms of the overall development of the Yangtze River Delta, DE, NU, LE, and coupling coordination generally show a stable upward trend. Among the subsystems, the overall development of LE is in the lead, and there is a gradual convergence trend among the subsystems over time. In terms of coupling coordination, the coupling degree is always greater than the coupling coordination degree and the development index of each subsystem, indicating that the connection among the subsystems is increasingly strengthened.

Second, in terms of spatial and temporal evolution, the subsystems and the coupling coordination show different evolutionary characteristics. The DE shows a tripod complexion of Shanghai, Hangzhou, and Nanjing, and the trend of polarization is obvious. The NU is distributed in a Z-shape with metropolitan areas of Hefei, Nanjing, Shanghai, Hangzhou, and Ningbo as the main axes. The LE shows a more balanced development trend during the study period, and gradually forms a spatial pattern of two horizontals of "Chizhou-Lishui" and "Nanjing-Zhoushan" and one vertical of "Quzhou-Nantong". The coupling coordination shows a typical pattern of "high in the east and low in the west" and has a tendency to spread horizontally from east to west.

Third, DE, NU, and LE in the Yangtze River Delta all have positive progressive effects and path-dependent characteristics, and the influence on themselves weakens over time. Among them, the self-reinforcing effect of DE lasts the longest. The DE has a lasting and strong driving effect on NU and LE, but the demand stimulation of NU and LE for DE still needs to be strengthened. The interaction between NU and LE is basically insignificant, which reflects the inherent land contradiction between urban development and land conservation.

Fourth, there are significant spatial effects of the DE, NU, LE, and the coupled coordination degree in the Yangtze River Delta, where the DE shows an obvious "siphon effect", which attracts the resource elements of surrounding cities and has adverse effects on their coordinated development. The NU and LE both have significant spatial spillover effects and can promote the coordinated development of neighboring cities.

### 6.2. Related Suggestions

Based on the above conclusions, this paper puts forward the following suggestions:

First, the administrators of the city should adhere to the concept of sustainable and coordinated development and perfect the integration mechanism of DE, NU, and LE. At present, differences in the development of subsystems still exist. Therefore, the admin-

istrators must fully understand the development characteristics of each subsystem and always carry out the concept of sustainable development. The old route of construction at the expense of the ecological environment should be discarded. The holistic view and the inner coupling mechanism of the subsystems should be established and clarified; thus, coordinating the implementation from the perspective of the whole area together with building to closely integrate the development of the DE, NU, and LE with the high-quality development of the Yangtze River Delta.

Second, each city should actively explore differentiated development paths and accelerate the integration of digital technology with economic and social development. First of all, Shanghai, Hangzhou, and Nanjing should increase investment in the development of the digital economy and speed up the promotion of the digital economy from polarization to diffusion. At the same time, local outstanding high-tech enterprises and Internet enterprises should play a positive role in achievement of transformation. The digital advantages of big cities can be converted into digital power for regional common development through knowledge diffusion and result sharing. Secondly, for the cities in Anhui Province with a lower level of coordinated development, on the one hand, they should seize the opportunity of the integrated development of the Yangtze River Delta region. Forming a synergistic development in economy and industry by the enhancement of collaboration and exchange with Jiangsu, Zhejiang, and Shanghai. A green development pattern should be built through the common protection, common monitoring, and common governance in the ecological field. On the other hand, cities in Anhui Province should focus on strengthening their development foundation; specially to strengthen the construction of northern Anhui to avoid the development gap between the north and the south continuing to expand. The advantages in agriculture, manufacturing, and energy industry should be emphasized. Through the application of digital technology, empowering agricultural production, accelerating the transformation of manufacturing, and improving the efficiency of energy create typical farms and factories for the Yangtze River Delta. Finally, for some ecological cities with a slightly weaker economic base such as Huangshan, Zhoushan, Lishui, and Huzhou, the current ecological and resource advantages are utilized fully to further solidify the ecological base of the Yangtze River Delta. The establishment of green ecological cities can enable ecological welfare to cover more areas and benefit more people through the externality of ecological environment and urban development.

**Author Contributions:** Conceptualization, R.D. and L.D.; data curation, J.Z. and L.P.; formal analysis, S.S.; visualization, J.F.; writing—original draft preparation, Y.Z.; writing—review and editing, Y.Z. and R.D. All authors have read and agreed to the published version of the manuscript.

**Funding:** This work is sponsored by the Innovation Exploration and Academic Seeding Project of Guizhou University of Finance and Economics (2022XSXMA04).

**Data Availability Statement:** All data used in this paper are available in the city almanac, local statistical almanac, and EPS database. The PM2.5 data are obtained from the Surface PM2.5 Dataset of the Atmospheric composition Analysis Group at Washington University in St. Louis. The Digital Finance Inclusion Index can be found on the website (https://www.dfor.org.cn/, accessed on 19 January 2023).

**Acknowledgments:** The authors are grateful to the statisticians for providing the data and to the editors and anonymous reviewers for their suggestions and comments.

**Conflicts of Interest:** The authors declare no conflict of interest.

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
