# Peer review of "Spatial and Temporal Interaction Coupling of Digital Economy, New-Type Urbanization and Land Ecology and Spatial Effects Identification: A Study of the Yangtze River Delta"

_land, doi:10.3390/land12030677_

Round 1

Reviewer 1 Report

General comments.

1.     This manuscript explores important and useful thematic areas for cities analysing big data in innovative ways. However, while the manuscript has shown, as indicated in the conclusion, that the interaction among city sub-systems of digital economy (DE), new type urbanization (NU) and land ecology (LE) has strengthened over time, there are major challenges in the manuscript. The lack of clarity in the introduction and detailing of the theory about what coupling and coordination means, as adapted from physics, and applied here to cities, leaves the reader continually debating whether the evidence presented is more about correlation rather than cause and effect. The material (e.g Figure 50) and the complexity of what is being analyzed (e.g. cities) points towards correlation. The statement in line 612 that the Grange causality test indicates that there is a two-way causal relationship among DE, NU and LE is not explained, described or proven across the manuscript. By contrast, the conclusions of the manuscript are strong – they indicate that the ‘inter-relationship among the sub-systems is getting stronger’ - and the approach of the manuscript should reflect these conclusions better.

2.     Introduction. For an international audience it is suggested that the term land ecology is described more and shown how it links to more commonly used international terms such as ecological system, natural resources, land restoration, ecosystem services etc.

3.     The Introduction should end with a summary of the different sections in the manuscript to guide the reader.

4.     The structuring of the manuscript is generally good.

5.     The Coupling Mechanism section 2. Figure 1 does not explain the mechanisms and the accompanying text does not match or explain the figure. Also, a summary is needed at the end of the section indicating what variables compose the coupling mechanism especially as this is part of the mathematical formula used in the paper in later sections. This section should come in 3. Study Area, Materials and Methods but before 3.2.2.

6.     The choice and interpretation of variables is absolutely critical in reaching conclusions that reflect reality. For e.g. land consumption/soil sealing OR land restoration off a baseline is not identified as a variable under LE. This is a significant variable given the fast pace of growth and sprawl on Chinese cities and the goal of ecological civilization and the work China is undertaking to restore land. And given the final conclusions on the contestation over the use of land being a key variable.

7.     Why are the changes described as cyclical? (e.g.line 445).

8.     Section 6.1 of the conclusions should be deleted as it introduces new material and conclusions should only focus on what has been discussed in the manuscript above.

9.     English edit required. For e.g. many sentences are too long (lines 129-135 is one sentence). Lots of acronyms which need to be spelt out (e.g. SDM, SAR, SEM).

Specific comments.

1.     Lines 74-77 Unclear why this is mentioned. Is this something to do with consumer behaviour?

2.     Lines 97-103 needs clarification – need to make clear how rural negative impacts like chemical fertiliser and pesticides are an indicator of land pressure

3.     Line 129 what is the PVAR model?

4.     Line 138 – is this supposed to be possible coordinated development?

5.     Line 141 the Introduction indicates that it is about coordination between DE, NU and LE. But line 141 only talks about urban and land – this is not the same.

6.     Lines 171-173 needs a reference otherwise is an unproven assertion.

7.     Line 210 not just yuan but also give USD for international audience

8.     Line 268 – as this is a key statement it needs to be referenced or evidenced produced in the paper

9.     Line 840 -what is D.T.? Complete reference needed

Author Response

Dear Editor and Reviewers,

Thank you very much for giving us an opportunity to revise our manuscript. We sincerely appreciate editors and reviewers for the letter and the reviewers’ comments concerning our manuscript (land-2253586).

Those comments are all valuable and very helpful for revising and improving our manuscript, as well as the important guiding significance to our researches. We have studied reviewers’ comments carefully and have made related corrections on the original manuscript. In addition to the reviewers’ suggestions, we re-checked the manuscript and improved the writing quality. All revised parts are marked in red in the revised manuscript. The specific revisions, responses and explanations are summarized as following. Please refer to the corresponding attachment for details.

We look forward to your response. Thank you very much.

Sincerely,

Rui Ding

Reviewer 2 Report

The paper seems to be well prepared to report what has happened in the study area. To check it out in terms of "balanced development", those authors employed three categorical concept of DE, NU and LE. Those conceptual variables includes more detailed variables containing a variety of data provided by the city government. The conclusion also seems to be fit enough to justify the starting hypothesis and questions. 

Generally speaking, the work seems to be progressed in a textbook style with solid model specifications. It is hard to argue against the logic and process of research work with regard to the modelling, data collection and estimation process. The paper surely showed that what has been happened in terms of sustainable development in the area. As a policy maker's view, however, the work seems to reveal a severe weak point. The role of government, both central and local government, has been overlooked. They may find the different strategies and approaches employed by those local governments. They may compare the different policies made by at least three governments of Shanghai, Hangzhou and Nanjing in the future. Exploring the cause and effect is more important than the description and explanation in the research community though those works are sufficiently well done. 

Author Response

(The authors gave the same response as above.)
